# Incidence of Asymptomatic *Shigella* Infection and Association with the Composite Index of Anthropometric Failure among Children Aged 1–24 Months in Low-Resource Settings

**DOI:** 10.3390/life12050607

**Published:** 2022-04-19

**Authors:** Sabiha Nasrin, Md. Ahshanul Haque, Parag Palit, Rina Das, Mustafa Mahfuz, Abu S. G. Faruque, Tahmeed Ahmed

**Affiliations:** Nutrition and Clinical Services Division, icddr,b, 68 Shaheed Tajuddin Ahmed Sharani, Dhaka 1212, Bangladesh; sabiha.nasrin@icddrb.org (S.N.); parag.palit@icddrb.org (P.P.); rina.das@icddrb.org (R.D.); mustafa@icddrb.org (M.M.); gfaruque@icddrb.org (A.S.G.F.); tahmeed@icddrb.org (T.A.)

**Keywords:** asymptomatic *Shigella* infection, childhood malnutrition, composite index of anthropometric failure (CIAF), MAL-ED

## Abstract

Asymptomatic or subclinical infection by diarrheal enteropathogens during childhood has been linked to poor health and nutritional outcomes. In this study, we aimed to assess the impact of asymptomatic *Shigella* infection on different forms of childhood malnutrition including the composite index of anthropometric failure (CIAF). We used data from 1715 children enrolled in the multi-country birth cohort study, MAL-ED, from November 2009 to February 2012. Monthly non-diarrheal stools were collected and assessed using TaqMan Array Cards (TAC). Poisson regression was used to calculate incidence rates of asymptomatic *Shigella* infection. Generalized estimating equations (GEE) were used to assess the association between asymptomatic *Shigella* infection and nutritional indicators after adjusting for relevant covariates. Incidence rates per 100 child-months were higher in Tanzania, Bangladesh and Peru. Overall, after adjusting for relevant covariates, asymptomatic *Shigella* infection was significantly associated with stunting (aOR 1.60; 95% CI: 1.50, 1.70), wasting (aOR 1.26; 95% CI: 1.09, 1.46), underweight (aOR 1.45; 95% CI: 1.35, 1.56), and CIAF (aOR 1.55; 95% CI: 1.46, 1.65) in all the study sites except for Brazil. The high incidence rates of asymptomatic *Shigella* infection underscore the immediate need for *Shigella* vaccines to avert the long-term sequelae involving childhood growth.

## 1. Introduction

Globally, *Shigella* spp. has been identified as the second leading pathogen causing diarrheal mortality, accounting for 13.2% of all diarrhea-related death [1,2,3]. However, children under five years old from low- and middle-income countries (LMICs) bear the brunt of the burden, with approximately 90 % of diarrhea-related deaths in children occurring in sub-Saharan Africa and south Asia [1,3,4]. Members of *Shigella* spp. belong to the enterobacteriacae family and are Gram-negative, non-motile bacilli [1]. The clinical manifestations of shigellosis in humans are attributed to four distinct species of the *Shigella* genus, namely: *S. dysenteriae, S. flexneri, S. boydii and S. sonnei* with multiple serotypes [2,5]. Repeated infection and diarrhea have also been reported [4]. *Shigella* infections cause population-level stunting and increased inflammatory markers, which may have long-term negative consequences on the cellular architecture of gastrointestinal tissues and cognitive development, as well as affecting the efficiency of mucosal vaccines [6]. While only a subset of individuals develops symptomatic disease, even in the absence of overt diarrhea, enteric infections are common in low-resource settings, contribute to environmental enteropathy, and can cause or exacerbate growth faltering, and decline in vaccine response [6]. The prevalence of asymptomatic *Shigella* spp. has been observed to vary from 4.9% to 17.8% [1] and was found to be associated with impaired linear growth during childhood [1]. Asymptomatic carriage of *Shigella* is regarded to be critical for the organism’s survival and resultant disease transmission in the community [7,8]. Furthermore, asymptomatic carriers may play a dangerous role in the transmission of multidrug-resistant *Shigella* spp. with the presence of potential virulence genes in diarrhea endemic areas [7].

One of the major contributors to the burden of diarrhea, and other related infectious diseases, is malnutrition [9]. It has been estimated that globally 19% (110 million) of children under five are moderately to severely underweight and 30% (170 million) are moderately to severely stunted [10]. Several studies have shown a causal relationship with childhood undernutrition being associated with an increased risk of morbidity and mortality [10], poor cognitive development in later childhood and development of chronic diseases in adulthood [11,12]. However, most of the studies have reported the data of stunting (height/length for age z score (HAZ/LAZ) <-2), wasting (weight for height/length (WHZ/WLZ) <-2), and underweight (weight for age (WAZ) <-2) separately [13]. Children who are underweight may have wasting or stunting, and some may experience all three types of anthropometric failure [14]. As a result, when numerous anthropometric failures exist, traditional indicators used to assess children’s nutritional status tend to underestimate overall undernutrition [14,15]. Peter Svedberg developed the composite index of anthropometric failure (CIAF) in 2000, which provides six independent measurements of undernutrition using the traditional nutritional indicators, and the combined values of these indicators measure the cumulative burden of childhood undernutrition [13,14,15]. Studies investigating composite anthropometric failures using these conventional indicators, among children who are asymptomatic carriers of *Shigella* infection, are uncommon.

Vaccines are one of the most effective ways to prevent illnesses, and *Shigella* vaccines are now being tested in clinical trials [16,17]. A multi-centric study reported that 85% of cases occurring in LMIC could be attributable to *S. flexneri* 2a, 3a and 6, together with *S. sonnei*, and, therefore, a quadrivalent vaccine targeting these strains is expected to provide significant protection in endemic regions [18]. Serotype-based vaccines include conjugate vaccines, carbohydrate vaccines, and live-attenuated or killed whole-cell vaccines [19,20,21,22]. However, due to the lack of ideal animal models or low/serotype-specific protection, no *Shigella* vaccine has reached the stage of commercialization to date [23].

In this study, we estimated the incidence of, and the childhood malnutrition associated with, asymptomatic/subclinical *Shigella* infections among children of less than two years enrolled in the MAL-ED multi-country birth cohort study and have attempted to establish a possible association between asymptomatic *Shigella* infections and composite anthropometric failure during early childhood.

## 2. Materials and Methods

### 2.1. Study Design and Participants

The Etiology, Risk Factors, and Interactions of Enteric Infections and Malnutrition and the Consequences for Child Health and Development Project (MAL-ED) was a multi-country birth cohort study conducted in eight sites: Dhaka (Bangladesh), Vellore (India), Bhakta pur (Nepal), Naushero Feroze (Pakistan), Venda (South Africa), Haydom (Tanzania), Fortaleza (Brazil), and Loreto (Peru). Details of the study have been previously described elsewhere [24]. Briefly, from November 2009 to February 2012, children were recruited within 17 days of birth if maternal age was 16 years or more, their family intended to remain in the area for more than 6 months, birthweight was more than 1500 grams, the child was not diagnosed with chronic/congenital anomaly, and was from a singleton pregnancy of the mother, and their siblings were not enrolled in the study. After obtaining written informed consent from the parents or legal guardian, the child was enrolled in the study [24]. All sites used identical standardized protocol for data collection.

### 2.2. Data Collection

Participants were followed from enrolment after birth until the age of two years. During enrolment, a standardized questionnaire was used to collect socio-demographic data [24]. The indicators water/sanitation, assets, maternal education, and income (WAMI) index, were used to calculate the socio-economic status (SES). SES of the families was measured at six, twelve, eighteen, and twenty-four months [24,25]. World Health Organization (WHO) recommendations were used to describe improved water and sanitation [26]. Filtering, boiling, or adding bleach to drinking water were considered as treatments. Monthly anthropometric measurements and vaccination records were reviewed. Fieldworkers visited households twice a week to collect information on illness and child feeding practices [27]. Monthly, non-diarrheal stool-samples were collected and transported to designated laboratories where these were processed following harmonized protocols, which were followed across all study sites [28].

### 2.3. Assessment of Nutritional Status

The nutritional status of the children was determined by field staff measuring their weight and length. Standard scales were used to measure the anthropometry of the children (seca gmbh & co. kg., Hamburg, Germany). Measurements were taken each month at a specific time, preferably in the morning, with only the most basic clothing and no shoes on. The 2006 WHO child growth criteria were used to compute weight-for-age (WAZ) and length-for-age (LAZ) z-scores [29]. 

### 2.4. Laboratory Testing

Field workers collected monthly non-diarrheal stool samples without a fixative, and raw stool aliquots were stored at −80 °C before nucleic acid extraction. All laboratory testing was completed at site-specific laboratories [30,31]. The QIAmp Fast DNA Stool Mini Kit (Qiagen, Hilden, Germany) was used to extract total nucleic acid from stool specimens from children who had completed two years of follow-up, as previously described [32]. The efficacy of extraction and amplification was assessed using extrinsic controls, such as phocine herpesvirus (PhHV) and bacteriophage MS2. Quantitative polymerase chain reaction (PCR) with custom-designed TaqMan Array Cards was used to identify 29 enteropathogens utilizing the AgPath One Step realtime PCR kit (ThermoFisher, Waltham, MA, USA), as described elsewhere [33,34,35]. *Shigella* spp. were detected using primer sets specific for the ipaH gene [11,33], and a cycle of threshold (Ct) value of less than 35 (Ct < 35) was use as the cut-off [11,36]. 

### 2.5. Statistical Analysis

Line graphs were used to visualize the outcome variables as well as asymptomatic *Shigella* infection by months. Descriptive statistics, such as proportion, mean and standard deviation, were used to summarize the baseline characteristics. Poisson regression was used to calculate incidence rates of asymptomatic infection by *Shigella*. Generalized estimating equations (GEE) were used to assess the association between asymptomatic *Shigella* infection and nutritional indicators after adjusting for relevant covariates. Relevant covariates, such as sex, WAMI index, maternal height, site and less than three alive children, were adjusted in the final model. Child age in months was adjusted in the GEE model as a time variable. STATA Version 15 (Stata Corp; College Station, TX, USA) was used for all analyses.

### 2.6. Ethics Statement

The study was approved by the Institutional Review Board for Health Sciences Research, University of Virginia, USA, as well as the respective governmental, local institutional, and collaborating institutional ethical review committees at each site [24].

## 3. Results

### 3.1. General Characteristics

A total of 1715 children were available with two years of follow up. Due to bias observed in a subset of length measurements at this location, children from the Pakistan site (*n* = 246) were omitted in this analysis. The study participants’ demographic characteristics are shown in Table 1. The site-specific prevalence of asymptomatic *Shigella* infection by follow-up is presented in Figure 1.

The prevalence of asymptomatic *Shigella* infection at the Bangladesh, Peru and Tanzania sites was greater than that from other sites. The lowest incidence of asymptomatic *Shigella* infection at all the time points during the study period was found in Brazil. The site-specific prevalence of stunting, wasting, underweight and CIAF by follow-up are shown in Figure 2. The prevalence of stunting, as well as of CIAF, were found to be increasing over time in Tanzania in comparison to other countries. Underweight and wasting was found to be the highest in India.

### 3.2. Incidence Rate of Asymptomatic Shigella Infection

Table 2 illustrates the site-specific incidence rate (IR) of asymptomatic *Shigella* infection. The overall incidence rate of asymptomatic *Shigella* infection per 100 child-months was 10.78% (95% CI: 10.42, 11.16). Tanzania (IR: 17.80; 95% CI: 16.57, 19.11) had the highest incidence rate followed by Peru (IR: 13.63; 95% CI: 12.56, 14.79) and Bangladesh (IR: 13.06; 95% CI: 12.02, 14.18). Compared to Bangladesh, after adjusting for sex, WAMI index, maternal height, site and less than three alive children, the incidence rate ratios (IRR) of asymptomatic *Shigella* infection in Brazil (aIRR: 0.49; 95% CI: 0.39, 0.60), India (aIRR: 0.91; 95% CI: 0.81, 1.02), Nepal (aIRR: 0.49; 95% CI: 0.42, 0.57), and South Africa (aIRR: 0.64; 95% CI: 0.55, 0.76) were significantly lower than those at other study sites. On the other hand, infection in Tanzania (aIRR: 1.07; 95% CI: 0.91, 1.25) and Peru (aIRR: 1.04; 95% CI: 0.92, 1.16) was higher but not at a statistically significant level.

### 3.3. Association between Asymptomatic Shigella Infection and Childhood Malnutrition

Table 3 illustrates the site-specific association between asymptomatic *Shigella* infection and different forms of childhood malnutrition. Across all the study sites, after adjusting for a number of covariates, namely, sex, WAMI index, maternal height, site and less than three alive children, asymptomatic *Shigella* infection was significantly associated with stunting (aOR 1.60; 95% CI: 1.50, 1.70), wasting (aOR 1.26; 95% CI: 1.09, 1.46), underweight (aOR 1.45; 95% CI: 1.35, 1.56), and CIAF (aOR 1.55; 95% CI: 1.46, 1.65). However, if we consider the site-specific strengths of association, asymptomatic *Shigella* infection was not significantly associated with any forms of malnutrition in Brazil. In the case of wasting, no significant association was found in India, Nepal, Peru, and South Africa. 

## 4. Discussion

Our findings demonstrated a disparity in the prevalence of asymptomatic *Shigella* infections across the sites, with incidence rates being greater in south Asian sites, Tanzania, and Peru than in other study sites. Several epidemiological studies on infectious diseases have demonstrated the importance of regional differences in disease risk and burden [37,38]. In our study, asymptomatic *Shigella* infection was associated with all forms of conventional indicators of malnutrition, including CIAF in children under the age of two years. In GEMS, a large study of moderate-to-severe diarrhea in seven sites in Africa and Asia, asymptomatic *Shigella* prevalence determined by qPCR was 27% among recruited controls without diarrhea in the second year of life [6]. Furthermore, asymptomatic infection with enteropathogens was strongly associated with linear growth faltering in childhood [6,39].

Currently, there is no vaccine against *Shigella* infection [16,17,40]. Based on the severity, disease burden, and emergence of antimicrobial resistance, *Shigella* has long been a priority for the WHO and other international organizations and the potential to prevent the disease in children would be a huge scientific breakthrough with substantial public health implications [17,18]. McQuade et al. reported that when the latest technique (qPCR) was used compared to culture, there was an 11-fold increase in *Shigella* detection compared to previous reports [1,16,17]. In this study, it was reported that traditional culture methods may have missed the majority of *Shigella* cases; the lowest sensitivity was found in young children, who are the most vulnerable to adverse outcomes [1]. The high incidence rate of asymptomatic infection in this study, as well as a putative link to growth outcomes, could increase the potential impact of a *Shigella* vaccine, especially in LMICs.

A previous study compared the antibiotic resistance and virulence gene profiles of *Shigella* spp. isolated from diarrheal and asymptomatic children under five years old and found minor variations in their phenotypic and genetic characteristics [7]. Resistance gene markers were found in mobile genetic elements of *Shigella* spp. isolated from controls, indicating that these organisms are more adapted to antibiotic pressure [7]. Apart from the invasive plasmid antigen H-encoding gene (ipaH), other virulence genes, including virF, sat, setA, setB, sen, and ial, were found in asymptomatic *Shigella* positive controls compared to *Shigella* isolates from symptomatic cases [7]. Although virulence markers have been found in those organisms isolated from asymptomatic children, the exact mechanisms of subclinical infection remain unknown [7]. Malnutrition, immune deficiency, poor hygiene, and prolonged excretion of the pathogen after the disease are thus the factors that contribute to asymptomatic *Shigella* spp. carriage [1,7,41,42].

Furthermore, the nutritional status in the first two years of life is considered to represent a “critical window” for development and growth of a child and has long-term effects persisting up to adulthood [43]. Malnutrition is synergistically associated with shigellosis [40]. Shigellosis causes a protein-losing enteropathy that can aggravate malnutrition and lead to adverse outcomes [40,44,45]. Shigellosis causes profuse loss of blood from the infected colon that continues even after the elimination of the pathogen [44,46]. Considerable loss of serum protein from the ulcerated colon is a cause for hypoproteinemia [47]. In our study we found an increasing prevalence over time with increasing age. A similar trend in age was found in a previous study [1]. Exclusive breastfeeding was a significant protective risk factor for *Shigella* infection [1]. As a child grows older and receives complementary feeding, or begins to eat the family diet, they are more likely to become infected with *Shigella*. Person-to-person spread is common among children who are mobile but have not yet developed adequate hygienic practices to avoid transmission. 

In our study, we observed a significant association between asymptomatic *Shigella* infection and CIAF in all countries except Brazil. To the best of our knowledge, this is the first study that has exhibited an association between asymptomatic *Shigella* infection and CIAF during early childhood. Previous studies have suggested that all grades of anthropometric deficits amplified the risks of death from diarrheal diseases and other infections [10]. According to a systematic review undertaken in developing countries, children who fail all three anthropometric assessments have a 12-fold increased risk of mortality [48]. In our study, Tanzania (50.7%) had the highest prevalence of CIAF, followed by India (43.1%) and Bangladesh (37.7%), which is in line with estimates reported by studies conducted in India and Bangladesh [13,14]. However, one study using the Tanzania Demographic and Health Survey found a significant downward trend in CIAF prevalence from 50% in 1991 to 38.2% in 2015, which was a lower level compared to our study [49].

The association between asymptomatic *Shigella* infection with stunting was highest in Bangladesh compared to other countries. Stunting and CIAF were the highest in Tanzania. Stunting is one of the most prevalent forms of chronic childhood undernutrition. In 2019, the global prevalence of stunting was 21.4%, whereas in Bangladesh, it was 36% [50,51], and in Tanzania, the national average of prevalence of stunting was 34% [50,52]. Numerous studies have found negative relations between stunting and child development [50]. The current rate of reduction in stunting in Bangladesh to meet the World Health Assembly’s aim of a 40% reduction in stunting levels by 2025 is insufficient [51]; the present annual rate of decrease in prevalence of stunting of 2.7% must be increased to 3.3% [51]. In Tanzania, the average prevalence of stunting has declined from almost 50% in 1992 to 34% in 2015, with stunting affecting one out of every three children under the age of five [52].

In Brazil, asymptomatic *Shigella* infection was not associated with any form of childhood malnutrition including the CIAF. According to an earlier study, the prevalence of stunting and undernutrition among children in Brazil between 1974 and 2009 dropped considerably [53]. Overweight and obesity rates, on the other hand, rose over the same period [53]. Another Brazilian study, conducted in Fortaleza, CE, Brazil, from 19 August 2010 to 30 September 2013, reported that, among children who attended the Institute for the Promotion of Nutrition and Human Development clinic for nutritional counselling, enteropathogenic *E. coli* (EPEC), enteroinvasive *E. coli* (EIEC), *Giardia* spp., enteroaggregative *E.coli* (EAEC), and *C. jejuni/coli* were the most prevalent pathogens in stool samples from both well-nourished and malnourished children [54]. 

The strengths of our study include the use of individual-level data from a large multi-country birth cohort [24]. We used the customized multiplex qPCR platform, known as TaqMan Array Cards, for the detection of *Shigella* infection. A limitation of the study is that the ipaH gene is associated with both *Shigella* and enteroinvasive *E. coli*. Previous speciation and metagenomic research, on the other hand, backs up the interpretation of these detections as *Shigella* [1]. For simplicity we have also used this only for *Shigella.* There could also be residual confounding not addressed by the covariates in the models that lead to an association that is not in fact causal.

## 5. Conclusions

For decades, the discovery and approval of a safe and highly effective *Shigella* vaccine has been a top priority in international public health communities, and it would be a significant scientific accomplishment. The high incidence of asymptomatic infection in this study, along with a possible link to growth outcomes, could boost *Shigella* vaccine’s potential impact, particularly in low- and middle-income countries where vaccination could prevent infection. The planned study will contribute to a better knowledge of the impact of pathogenesis of *Shigell*a in relation to human health, as *Shigella* vaccines are currently being tested in clinical trials. However, in-depth studies are needed to assess all aspects of the path from enteric infection to long-term morbidity, including environmental enteric dysfunction, malnutrition, growth faltering, and cognition.

## Figures and Tables

**Figure 1 life-12-00607-f001:**
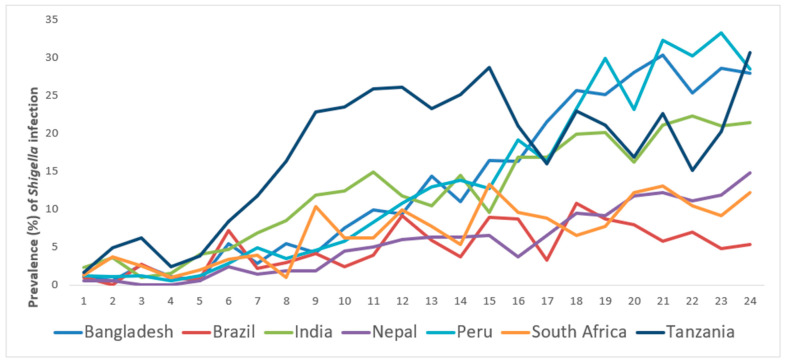
Site specific prevalence of asymptomatic *Shigella* infection of MAL-ED study children from November 2009 to February 2012 by follow-up.

**Figure 2 life-12-00607-f002:**
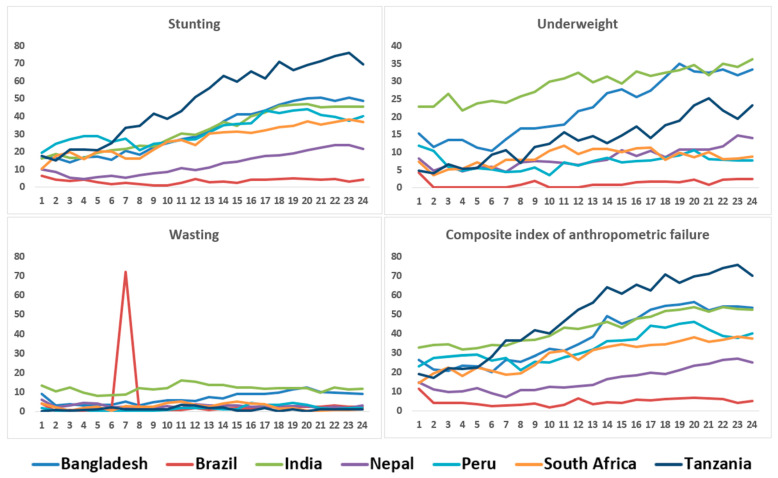
Site-specific prevalence (%) of stunting, wasting, underweight and composite index of anthropometry failure by follow-up.

**Table 1 life-12-00607-t001:** General characteristics of MAL-ED study populations from November 2009 to February 2012 (n = 1715).

Characteristics, n (%)	Bangladesh	Brazil	India	Nepal	Peru	Pakistan	South Africa	Tanzania	Overall
Male sex	108 (51.4)	89 (53.9)	105 (46.3)	122 (53.7)	105 (54.1)	120 (48.8)	120 (50.6)	105 (50.2)	874 (51.0)
Birth weight (kg) ^†^	2.8 ± 0.4	3.4 ± 0.5	2.9 ± 0.4	3 ± 0.4	3.1 ± 0.4	2.7 ± 0.4	3.2 ± 0.5	3.2 ± 0.5	3.0 ± 0.5
Days of exclusive breastfeeding ^†^	143.2 ± 42.7	93.7 ± 57.8	105.4 ± 42.9	92.5 ± 54.5	89.5 ± 61.3	19.9 ± 22.7	38.6 ± 26.3	62.2 ± 35	78.6 ± 57.7
Weight for age z-score at enrolment ^†^	−1.3 ± 0.9	−0.2 ± 1	−1.3 ± 1	−0.9 ± 1	−0.6 ± 0.9	−1.4 ± 1	−0.4 ± 1	−0.1 ± 0.9	−0.8 ± 1.1
Length for age z-score at enrolment ^†^	−0.96 ± 1	−0.8 ± 1.1	−1 ± 1.1	−0.7 ± 1	−0.9 ± 1	−1.3 ± 1.1	−0.7 ± 1	−1 ± 1.1	−0.9 ± 1.1
Length for age z-score at 24 months ^†^	−2.0 ± 0.9	0 ± 1.1	−1.9 ± 1	−1.3 ± 0.9	−1.9 ± 0.9	N/A	−1.7 ± 1.1	−2.7 ± 1	−1.7 ± 1.2
Maternal age (years) ^†^	25.0 ± 5.0	25.4 ± 5.6	23.9 ± 4.2	26.6 ± 3.7	24.8 ± 6.3	28.1 ± 5.9	27 ± 7.2	29.1 ± 6.5	26.3 ± 5.9
Maternal weight (kg) ^†^	49.7 ± 8.5	62 ± 11.5	50.3 ± 9.3	56.2 ± 8.3	56.3 ± 9.6	50.7 ± 9.6	68 ± 15.3	55.7 ± 8.8	55.9 ± 12
Maternal height (cm) ^†^	149.0 ± 5.0	155.1 ± 6.7	151.1 ± 5.2	149.7 ± 5.3	150.2 ± 5.5	153.4 ± 5.7	158.7 ± 6.6	155.9 ± 5.9	152.9 ± 6.6
Maternal educational level < 6 y	133 (63.3)	22 (13.3)	80 (35.2)	59 (26)	44 (22.7)	202 (82.1)	5 (2.1)	75 (35.9)	620 (36.2)
Mother has less than 3 alive children	160 (76.2)	113 (68.5)	157 (69.8)	199 (87.7)	111 (57.2)	105 (42.7)	141 (59.5)	58 (27.8)	1044 (61)
Routine treatment of drinking water	130 (61.9)	10 (6.1)	7 (3.1)	98 (43.2)	32 (16.5)	0 (0)	12 (5.1)	12 (5.7)	301 (17.6)
Improved drinking water source	210 (100)	165 (100)	227 (100)	227 (100)	184 (94.9)	246 (100)	196 (82.7)	89 (42.6)	1544 (90.0)
Improved floor	204 (97.1)	165 (100)	222 (97.8)	109 (48)	69 (35.6)	81 (32.9)	231 (97.5)	13 (6.2)	1094 (63.8)
Improved latrine	210 (100)	165 (100)	121 (53.3)	227 (100)	66 (34)	197 (80.1)	232 (97.9)	19 (9.1)	1237 (72.1)
Monthly income < $150	69 (32.9)	161 (97.6)	19 (8.4)	106 (46.7)	58 (29.9)	115 (46.8)	179 (75.5)	0 (0)	707 (41.2)

^†^ Mean ± Standard deviation.

**Table 2 life-12-00607-t002:** Site-specific incidence rate and incidence rate ratio compared with Bangladesh.

	Incidence Rate per 100 Child-Months (95% CI)	Adjusted Incidence Rate Ratio (95% CI) *	*p* Value
** *Shigella* **			
Overall	10.78 (10.42, 11.16)		
Bangladesh	13.06 (12.02, 14.18)	Reference	
Brazil	4.89 (4.15, 5.78)	0.49 (0.39, 0.60)	<0.001
India	12.41 (11.45, 13.46)	0.91 (0.81, 1.02)	0.136
Nepal	5.75 (5.12, 6.45)	0.49 (0.42, 0.57)	<0.001
Peru	13.63 (12.56, 14.79)	1.04 (0.92, 1.16)	0.549
South Africa	7.01 (6.28, 7.82)	0.64 (0.55, 0.76)	<0.001
Tanzania	17.80 (16.57, 19.11)	1.07 (0.91, 1.25)	0.379

* Adjusted for sex, WAMI Index (water/sanitation, assets, maternal education, and income), maternal height, mother has less than 3 alive children, and site for overall estimate.

**Table 3 life-12-00607-t003:** Site-specific strength of association between *Shigella* infection and child’s nutritional status.

	Unadjusted OR (95% CI)	*p* Value	Adjusted OR (95% CI)	*p* Value
**Stunting [LAZ<-2]**				
Overall	1.54 (1.45, 1.62)	<0.001	1.60 (1.50, 1.70)	<0.001
Bangladesh	1.97 (1.73, 2.24)	<0.001	2.09 (1.81, 2.41)	<0.001
Brazil	1.36 (0.69, 2.64)	0.368	1.33 (0.64, 2.79)	0.447
India	1.40 (1.23, 1.58)	<0.001	1.41 (1.24, 1.61)	<0.001
Nepal	1.54 (1.24, 1.91)	<0.001	1.62 (1.26, 2.08)	<0.001
Peru	1.44 (1.26, 1.65)	<0.001	1.50 (1.29, 1.73)	<0.001
South Africa	1.32 (1.11, 1.57)	0.002	1.35 (1.12, 1.63)	0.002
Tanzania	1.58 (1.40, 1.79)	<0.001	1.65 (1.44, 1.89)	<0.001
**Wasting [WLZ<-2]**				
Overall	1.26 (1.10, 1.44)	0.001	1.26 (1.09, 1.46)	0.002
Bangladesh	1.49 (1.15, 1.93)	0.003	1.50 (1.15, 1.95)	0.003
Brazil	0.54 (0.16, 1.83)	0.325	0.57 (0.14, 2.35)	0.433
India	1.15 (0.95, 1.40)	0.146	1.15 (0.95, 1.40)	0.143
Nepal	1.22 (0.69, 2.20)	0.487	1.24 (0.67, 2.29)	0.488
Peru	0.47 (0.19, 1.12)	0.088	0.45 (0.17, 1.19)	0.109
South Africa	1.53 (0.88, 2.66)	0.133	1.53 (0.88, 2.64)	0.129
Tanzania	2.06 (1.13, 3.77)	0.019	2.06 (1.12, 3.77)	0.020
**Underweight [WAZ<-2]**				
Overall	1.42 (1.33, 1.51)	<0.001	1.45 (1.35, 1.56)	<0.001
Bangladesh	1.81 (1.58, 2.07)	<0.001	1.90 (1.64, 2.20)	<0.001
Brazil	0.52 (0.07, 3.93)	0.529	0.56 (0.11, 2.85)	0.485
India	1.31 (1.16, 1.48)	<0.001	1.31 (1.16, 1.48)	<0.001
Nepal	1.63 (1.28, 2.08)	<0.001	1.71 (1.27, 2.30)	<0.001
Peru	1.06 (0.84, 1.33)	0.649	1.06 (0.82, 1.38)	0.646
South Africa	1.36 (1.06, 1.76)	0.016	1.38 (1.05, 1.80)	0.019
Tanzania	1.38 (1.17, 1.63)	<0.001	1.40 (1.18, 1.68)	<0.001
**Composite index of anthropometric failure [LAZ<-2 or WLZ<-2 or WAZ<-2]**				
Overall	1.49 (1.41, 1.58)	<0.001	1.55 (1.46, 1.65)	<0.001
Bangladesh	1.95 (1.71, 2.22)	<0.001	2.02 (1.77, 2.33)	<0.001
Brazil	1.05 (0.58, 1.87)	0.881	1.02 (0.58, 1.83)	0.924
India	1.31 (1.16, 1.48)	<0.001	1.32 (1.16, 1.50)	<0.001
Nepal	1.54 (1.25, 1.89)	<0.001	1.61 (1.27, 2.04)	<0.001
Peru	1.39 (1.22, 1.59)	<0.001	1.45 (1.25, 1.67)	<0.001
South Africa	1.33 (1.12, 1.58)	0.001	1.36 (1.13, 1.63)	0.001
Tanzania	1.54 (1.36, 1.75)	<0.001	1.60 (1.40, 1.84)	<0.001
**Stunting and underweight only [LAZ<-2 and WAZ<-2]**				
Overall	1.42 (1.30, 1.56)	<0.001	1.42 (1.29, 1.56)	<0.001
Bangladesh	1.73 (1.45, 2.07)	<0.001	1.79 (1.47, 2.17)	<0.001
Brazil	-	-	-	-
India	1.29 (1.08, 1.54)	0.005	1.27 (1.06, 1.52)	0.011
Nepal	1.54 (1.09, 2.16)	0.013	1.59 (1.08, 2.34)	0.019
Peru	1.26 (0.94, 1.68)	0.117	1.28 (0.94, 1.75)	0.117
South Africa	1.23 (0.87, 1.73)	0.238	1.24 (0.86, 1.77)	0.246
Tanzania	1.37 (1.15, 1.65)	0.001	1.40 (1.15, 1.69)	0.001

Adjusted in generalized linear model for sex, WAMI Index (water/sanitation, assets, maternal education, and income), maternal height, mother has less than 3 alive children and site for overall estimate. Dependent variables: stunting (LAZ<-2), wasting [WLZ<-2], underweight [WAZ<-2], composite index of anthropometric failure [LAZ<-2 or WLZ<-2 or WAZ<-2], and stunting and underweight only [LAZ<-2 and WAZ<-2]; Independent variables: asymptomatic *Shigella* infection.

## Data Availability

All relevant data, including personal data, is available upon request from the ClinEpiDB database (https://clinepidb.org/ce/app/record/dataset/DS_3dbf92dc05, accessed on 15 January 2021).

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
