# Peer review of "Incidence of Asymptomatic Shigella Infection and Association with the Composite Index of Anthropometric Failure among Children Aged 1–24 Months in Low-Resource Settings"

_life, 2022, doi:10.3390/life12050607_

Round 1

Reviewer 1 Report

I guess this paper was written some time ago. Not a single reference from 2021-2022. Please edit text fpr clarity and add latest stats about Shigella burden in infants.

No need of a separate heading for strengths and limitations. It falls under discussion.

Conclusion should be more critical and introduction should be more coherent and crisp.

Presentation of article is a major issue. Lots of good data butnot beautifully done. Please alter in light of recent literature.

Author Response

Point 1: I guess this paper was written some time ago. Not a single reference from 2021-2022. Please edit text for clarity and add latest stats about Shigella burden in infants.

Response 1: Thank you very much for your thoughtful evaluation and helpful advice. In the introduction, we have included recent statistics on Shigella infection in children. Also, we tried to improve the writing in light of recent literature.

Point 2: No need of a separate heading for strengths and limitations. It falls under discussion.

Response 2: Authors appreciate your suggestion. We have integrated the strengths and limitation section with the discussion section as per your suggestion (lines 162-170).

Point 3: Conclusion should be more critical and introduction should be more coherent and crisp.

Response 3: Thank you so much for your valuable opinion. We attempted to improve the writing by making the conclusion (lines 171-190) more critical and the introduction (lines 30-102) more cohesive and concise.

Point 4: Presentation of article is a major issue. Lots of good data but not beautifully done. Please alter in light of recent literature.

Response 4: Thank you so much for taking the time to provide your valuable opinion. In view of recent literature, we tried to improve the writing.

Reviewer 2 Report

This paper summarizes a secondary analysis of the MAL-ED cohort study conducted over a decade ago but leveraging more recent molecular diagnostics to better identify diarrheal pathogens, particularly Shigella. The authors use data on asymptomatic Shigella infection and growth outcomes to identify potential associations with asymptomatic Shigella infection and growth faltering, which has previously been noted. 

The paper is a useful contribution to articulating the potential value of a Shigella vaccine. The high incidence of asymptomatic infection coupled with a potential association with growth outcomes could increase Shigella vaccine's potential impact, if vaccination prevented infection. 

The paper would benefit from some general editing to address typos and incomplete sentences (e.g. page 1 line 43, page 2 line 64, page 11 line 122). Page 7 is awkwardly worded in the presentation of IRs. I am not sure that a site-to-site comparison is particularly useful beyond the site-specific incience rates, and it is unclear why Bangladesh is the reference country: why not the countries with the lowest incidence? The discussion section would benefit from relating to other literature on Shigella infection and growth faltering. Some relevant summary data on Shigella infection in non-diarrheal stools from the McQuade 2020 paper should be included, as they otherwise seem like omissions. Parts of the discussion were not easy to follow in terms of their significance/relevance to the findings of the paper, and key points could be summarized more succintly.

A few aspects of the methods would benefit from clarification: 1) did asymptomatic infections also require a Ct value of <35, or were higher values included? 2) at what visit relative to asymptomatic Shigella infection were the analyzed anthropometric measurements taken? 

Figures 1: Stratifying the data by region across different graphs made it more difficult to compare. Consider putting all on the same graph and using color or line style to differential regions. Across several sites there seems to be an increasing trend in asymptomatic infection over the 24 months of follow up. Is this a secular trend or age-related? It would be useful to comment on this in the results/discussion. If a vaccine intervention were effective against asymptomatic infection, what do your results suggest about a preferred schedule?

Figure 2: The y-axis should be consistent across Asian/African/South American graphs. Like Figure 1, I am not sure it is effective to split these apart. Furthermore, it would be nice if Figures 1 and 2 could be more directly compared so as to show visually the relationships between asymptomatic infection by study month and nutritional status measurements. My suggestion would be not to stratify by region and layer monthly prevalence, stunting, wasting, underweight, and CIAF into a multi-part, single figure with the months lined up across all.

For the limitations, I would add that there could be residual confounding not addressed by the covariates in the models that lead to an association that is not in fact causal.

Journals are missing from the citations on the reference list.

Author Response

This paper summarizes a secondary analysis of the MAL-ED cohort study conducted over a decade ago but leveraging more recent molecular diagnostics to better identify diarrheal pathogens, particularly Shigella. The authors use data on asymptomatic Shigella infection and growth outcomes to identify potential associations with asymptomatic Shigella infection and growth faltering, which has previously been noted. 

The paper is a useful contribution to articulating the potential value of a Shigella vaccine. The high incidence of asymptomatic infection coupled with a potential association with growth outcomes could increase Shigella vaccine's potential impact, if vaccination prevented infection. 

Point 1: The paper would benefit from some general editing to address typos and incomplete sentences (e.g. page 1 line 43, page 2 line 64, page 11 line 122).

Response 1: Thank you so much for your valuable opinion. We have corrected the lines (details below).

Page 1 line 43- Corrected now on page 2 line 51.

Page 2 line 64- was removed during revision

Page 11 line 122 -was removed as this line appeared to be unnecessary.

Point 2: Page 7 is awkwardly worded in the presentation of IRs.

Response 2: Thank you so much for the comment. We have edited this section to make it easy (Page-8, lines 18-31).

Point 3: I am not sure that a site-to-site comparison is particularly useful beyond the site-specific incience rates, and it is unclear why Bangladesh is the reference country: why not the countries with the lowest incidence?

Response 3: Thank you so much for taking the time to leave such an important feedback. As, we work in Bangladesh and intend to continue working on this topic in the future, we chose Bangladesh as the reference country to highlight.

Point 4: The discussion section would benefit from relating to other literature on Shigella infection and growth faltering. Some relevant summary data on Shigella infection in non-diarrheal stools from the McQuade 2020 paper should be included, as they otherwise seem like omissions. Parts of the discussion were not easy to follow in terms of their significance/relevance to the findings of the paper, and key points could be summarized more succintly.

Response 4: The authors agree that summary statistics from the McQuade 2020 study should be included. We rewrote the discussion section to make it more understandable.

Point 5: A few aspects of the methods would benefit from clarification: 1) did asymptomatic infections also require a Ct value of <35, or were higher values included? 2) at what visit relative to asymptomatic Shigella infection were the analyzed anthropometric measurements taken? 

Response 5: Thank you so much for the valuable comment. 1) Monthly, non-diarrheal stool-samples were collected and transported to designated laboratories (page 3, lines-127, 139). Quantitative PCR with custom-designed TaqMan Array Cards was used to identify 29 enteropathogens. Both symptomatic and asymptomatic stool samples Shigella pathogen were detected by Ct value <35 . We have mentioned this in the laboratory testing section (page 4, lines-142-152).

2) Monthly anthropometric measurements were taken. This has been clarified in the assessment of nutritional section (page 3, lines-125, 130).

Point 6: Figures 1: Stratifying the data by region across different graphs made it more difficult to compare. Consider putting all on the same graph and using color or line style to differential regions.

Response 6: We appreciate and accept your suggestion. To make it easier to compare, we edited figure 1 and made it a single figure (page 6). 

Point 7: Across several sites there seems to be an increasing trend in asymptomatic infection over the 24 months of follow up. Is this a secular trend or age-related? It would be useful to comment on this in the results/discussion. If a vaccine intervention were effective against asymptomatic infection, what do your results suggest about a preferred schedule?

Response 7: Thank you so much for the valuable comment. Information of age related increasing trend in asymptomatic Shigella infection was incorporated in the discussion section (page 11, line 105-109). Now it reads “In our study we found an increasing prevalence over the time with increasing age. Similar trend in age was found in previous study. Exclusive breastfeeding was a significant protective risk factor for Shigella infection. As a child grows older and receives complementary feeding or begins to eat the family diet, they are more likely to become infected with Shigella. Nonetheless, Pperson-to-person spread is common among children, who are mobile but have not yet developed hygienic practices ade-quately to avoid transmission”.

Point 8: Figure 2: The y-axis should be consistent across Asian/African/South American graphs. Like Figure 1, I am not sure it is effective to split these apart. Furthermore, it would be nice if Figures 1 and 2 could be more directly compared so as to show visually the relationships between asymptomatic infection by study month and nutritional status measurements. My suggestion would be not to stratify by region and layer monthly prevalence, stunting, wasting, underweight, and CIAF into a multi-part, single figure with the months lined up across all.

Response 8: Thank you so much for your kind review and valuable suggestion. As suggested we have simplified the figures 2 by monthly prevalence of stunting, wasting, underweight, and CIAF (page 7). 

Point 9: For the limitations, I would add that there could be residual confounding not addressed by the covariates in the models that lead to an association that is not in fact causal.

Response 9: We appreciate your useful input and have updated the limitation section to reflect it (page13, lines 169-170).

Point 10: Journals are missing from the citations on the reference list.

Response 10: Thank you very much for bringing this to our attention. The references have been updated, and the journal names have been added (page 14, lines 235-436).

Reviewer 3 Report

Life-1650141-peer-review-v1

Incidence of asymptomatic Shigella infection and association th the composite index of anthropometric failure among children aged 1-24 months in low-resource settings

The study aimed to assess the impact of asymptomatic Shigella infection on different forms of childhood malnutrition including the composite index of anthropometric failure (CIAF).

Comments

The proportion of Shigella infections that occur asymptomatically in young children has not been established. Inadequate access to food and infectious disease are the primary causes of childhood malnutrition.

The study revealed that asymptomatic Shigella infection was significantly associated with stunting, wasting, underweight  and CIAF in all the study sites except for Brazil.

I think  that this study is very interesting because epidemiological data, combined with new generation diagnostics, has highlighted a greater burden of Shigella disease than was previously estimated, which is not restricted to vulnerable populations in low-middle income countries.

The complex role of the gut microbiome in preventing and inducing such infections  could be explained different results in Brazil.

In my opinion the paper is well written with clear objective. The study  methodology  is adequate and the authors did an appreciable work.

I have not major or minor remarks

Author Response

The proportion of Shigella infections that occur asymptomatically in young children has not been established. Inadequate access to food and infectious disease are the primary causes of childhood malnutrition.

The study revealed that asymptomatic Shigella infection was significantly associated with stunting, wasting, underweight  and CIAF in all the study sites except for Brazil.

I think  that this study is very interesting because epidemiological data, combined with new generation diagnostics, has highlighted a greater burden of Shigella disease than was previously estimated, which is not restricted to vulnerable populations in low-middle income countries.

The complex role of the gut microbiome in preventing and inducing such infections  could be explained different results in Brazil.

In my opinion the paper is well written with clear objective. The study  methodology  is adequate and the authors did an appreciable work.

I have no major or minor remarks

Response 1: Thank you so much for your thoughtful evaluation and insightful remarks. Your feedback is much appreciated.

Round 2

Reviewer 1 Report

Acceptable in present form.